# Superplastic Deformation Mechanisms in Fine-Grained 2050 Al-Cu-Li Alloys

**DOI:** 10.3390/ma13122705

**Published:** 2020-06-14

**Authors:** Hongping Li, Xiaodong Liu, Quan Sun, Lingying Ye, Xinming Zhang

**Affiliations:** 1School of Materials Science and Engineering, Central South University, Changsha 410083, China; 193101043@csu.edu.cn (H.L.); liuxd87@csu.edu.cn (X.L.); 0607120910@csu.edu.cn (Q.S.); xmzhang@csu.edu.cn (X.Z.); 2Shanghai Aircraft Design and Research Institute of COMAC, Shanghai 200232, China; 3Key Laboratory of Nonferrous Materials, Ministry of Education, Central South University, Changsha 410083, China

**Keywords:** Al-Cu-Li alloys, superplasticity, creep diffusion, grain boundary sliding, focused ion beams

## Abstract

The deformation behavior and microstructural evolution of fine-grained 2050 alloys at elevated temperatures and slow strain rates were investigated. The results showed that significant dynamic anisotropic grain growth occurred at the primary stage of deformation. Insignificant dislocation activity, particle-free zones, and the complete progress of grain neighbor switching based on diffusion creep were observed during superplastic deformation. Quantitative calculation showed that diffusion creep was the dominant mechanism in the superplastic deformation process, and that grain boundary sliding was involved as a coordination mechanism. Surface studies indicated that the diffusional transport of materials was accomplished mostly through the grain boundary, and that the effect of the bulk diffusion was not significant.

## 1. Introduction

Al-Li alloy has a wide applicability in the field of aerospace because of its relatively low density, high elastic modulus, high specific strength, and excellent comprehensive properties [1,2,3]. Al-Li alloy is currently one of the best candidate alloys to replace the traditional 2xxx and 7xxx series of high-strength aluminum alloys [4]. As a new type of Al-Li alloy, Al-Cu-Li alloy not only has the characteristics of traditional Al-Li alloy, but also has excellent thermal stability and corrosion resistance [5]. However, the relatively low ductility at room temperature greatly limits the wide industrial application of Al-Li alloys. Superplastic forming provides an effective production process to solve this problem [4]. At present, the research on Al-Cu-Li alloy is mainly focused on corrosion properties, aging precipitation, mechanical properties, and the homogenization process [6,7,8,9,10], but there are few studies on superplastic forming.

Fine-grained polycrystalline metallic materials can be elongated by hundreds or even thousands of percent at elevated temperatures above 0.5T_m_ and relatively low strain rates, which is associated with the high sensitivity of the flow stress to the strain rate. This phenomenon was first discovered by Pearson in 1934 [11], when an elongation of 1950% was obtained in a Bi-Sn eutectic alloy. The ability to implement superplastic forming in the manufacturing of complex-shaped products has attracted a great deal of research interest since Backofen [12] first used superplastically formed components in 1964. Although many efforts have been made in the field of superplasticity in the past few decades, the understanding of the mechanism of superplastic deformation remains inadequate. It is currently accepted that grain boundary sliding (GBS) is the fundamental mechanism of superplastic deformation, and that diffusion creep (DC) and intragranular dislocation slip (IDS) coordinate grain boundary sliding. An important reason is that during superplastic deformation, even if the sample is stretched to a very high elongation, the grains tend to remain equiaxed [13], while neither diffusion creep nor intragranular dislocation slip are able to maintain the shape and size of the grain after superplastic deformation. Even in the idealized grain-neighbor switching model of Ashby–Verrall [14] based on diffusion creep, significant grain elongation still occurs after one cycle.

Although the view of grain boundary sliding as a function of the deformation mechanism of superplastic formation is very popular, there are many experimental observations that contradict it. For example, very weak grain boundary sliding and strong dislocation activity were observed during the superplastic deformation of an Al-Mg-Mn alloy, in which the contribution of intragranular dislocation slip to the whole deformation was close to 40% [15,16,17]. In studies of Al-Cu-Zr alloy by Bate et al. [18,19] and AA5083 by Todd et al. [20], it is believed that diffusion creep dominates the entire superplastic deformation process, and that the effect of grain boundary sliding and intragranular dislocation activity remains weak and negligible. Grain neighbor switching based on diffusional material transportation and grain boundary migration was indirectly observed through surface studies in AA5083 [20]. In these alloys, weak grain boundary sliding, significant dynamic grain growth, and dynamic recrystallization often occurred during superplastic deformation. Moreover, the transverse grain boundaries were formed by dislocation walls, and the grains were divided into several parts, which often occurred in elongated grains.

Surface studies have always been a common and effective method for the investigation of superplastic deformation mechanisms. Surface scratches have long been used as markers to show grain boundary offset during superplastic deformation [21,22]. However, in earlier studies, surface scratches were irregular and directionless, so it was impossible to quantitatively analyze intragranular strains. With the advent of focused ion beam technology, equidistant grids could be etched onto the surface of the samples to observe grain boundary offsets and strains inside the grain [23,24]. This accurate grid marking not only provided the possibility for further understanding of the superplastic deformation process, but also provided a means for the quantitative analysis of the contribution of various deformation mechanisms. Thus, the deformation behavior and microstructural evolution of the 2050 Al-Cu-Li alloy were investigated by combining surface and internal microstructure research, finally focusing on the quantitative analysis of the contribution of existing deformation mechanisms in the entire process of superplastic deformation. This study aims to provide experimental evidence for the superplastic forming of the 2050 Al-Cu-Li alloy, which will help in the optimization of the superplastic forming process.

## 2. Experimental Methods

### 2.1. Materials

Hot-rolled 2050 alloy with a thickness of 25.0 mm was used in this study (the composition is given in Table 1). The as-received plate was solution-treated at 520 °C for 3 h, then water cooled to room temperature. The plate was then subjected to a cold rolling pre-deformation with a thickness reduction of 40%, then overaged at 400 °C for 48 h, subsequently warm-rolled to 2.0 mm at 200 °C and recrystallized at 470 °C for 30 min in a molten salt bath. Through the thermo-mechanical processing, a fine-grained 2050 sheet ready for superplastic deformation was obtained, with an average grain size of 10.9 μm in the rolling direction and 8.4 μm in the transverse direction.

### 2.2. Tensile Test

All tensile tests were carried out on a RWS50 test machine (SINOTEST, Changchun, China). The samples for the tests were cut parallel to the rolling direction (RD) of the sheet and had a 10.0 mm parallel gauge length and 6.0 mm gauge width. The samples were tested at 490 °C and held at the test temperature for 20 min prior to being strained at the initial strain rate range of 1 × 10^−4^ s^−1^ to 1 × 10^−3^ s^−1^.

The samples for the surface study were mechanically polished using SiC papers, and finally using a polishing cloth with a diamond polishing paste. Then, square grids with two scales were milled into the polished surface using a Helios Nanolab 600i equipped with a focused ion beam (FIB) instrument (FEI, Hillsboro, OR, USA). The size of the coarse grid used to calculate the grain boundary sliding was 200 × 200 μm^2^, with a pitch of 20.0 μm and a nominal depth of 0.5 μm. The size of the fine grid used to calculate the intragranular deformation was 60 × 60 μm^2^, with a pitch of 3.0 μm and a nominal depth of 0.2 μm. The sample with the coarse and fine grids etched onto it by the FIB was gradually deformed to the true strain of 0.18, 0.36, and 0.54, respectively, at a temperature of 490 °C and an initial strain rate of 2 × 10^−4^ s^−1^. The other sample was deformed to a strain of 0.54 before polishing and grid etching, then the sample with the grids was further deformed to the true strain of 0.71, 0.89, and 1.09, respectively. The surface structure was recorded and analyzed in each true strain by a Zeiss EVO MA10 scanning electron microscope (SEM, ZEISS, Oberkochen, Germany).

The contribution of GBS was calculated by the offsets of transverse grid lines using the following formulas [25]:(1)$$\varepsilon_{\mathrm{GBS}}^{\mathrm{tr}}=\frac{\sum W_{i}}{l_{0}}$$
(2)$$\varepsilon^{\mathrm{tr}}=\frac{l_{f}-l_{0}}{l_{0}}$$
(3)$$\gamma_{\mathrm{GBS}}=\frac{\varepsilon_{\mathrm{GBS}}^{\mathrm{tr}}}{\varepsilon^{\mathrm{tr}}}=\frac{\sum W_{i}}{l_{f}-l_{0}}$$
where $\varepsilon_{\mathrm{GBS}}^{\mathrm{tr}}$ is the transverse strain induced by GBS; $\varepsilon^{\mathrm{tr}}$ is the whole transverse strain; $l_{0}$ and $l_{f}$ are the lengths of the transverse line of the coarse grid before and after the tensile test; $W_{i}$ is the length of overlap between the transverse grid lines; and $\gamma_{\mathrm{GBS}}$ is the contribution of GBS to the whole strain.

The contribution of intragranular deformation was calculated using the following formulas:(4)$$\varepsilon_{\mathrm{IDS}}=\frac{l_{f}^{f}-l_{0}^{f}}{l_{0}^{f}}$$
(5)$$\varepsilon =\frac{l_{f}^{c}-l_{0}^{c}}{l_{0}^{c}}$$
(6)$$\gamma_{\mathrm{IDS}}=\frac{\varepsilon_{\mathrm{IDS}}}{\varepsilon }$$
where $l_{0}^{f}$ and $l_{f}^{f}$ are the distances between the transverse lines of the fine grid within a grain, while $l_{0}^{c}$ and $l_{f}^{c}$ are the length of the longitudinal line of the coarse grid before and after deformation.

### 2.3. Microstructure Analysis

Microstructural observations of the deformed and static annealing samples were conducted using an Olympus-BX51M optical microscope (OM, Olympus, Tokyo, Japan). Specimens were prepared by mechanical polishing and anodic oxidization in a 3% HBF_4_ water solution at 20 V. Grain sizes in the rolling direction (RD) and the transverse direction (TD), as well as in the longitudinal and transverse planes, were calculated using the random secant method; more than 300 grains were measured in each direction. Electron back-scatter diffraction (EBSD) specimens were prepared by mechanical and electrolytic polishing in a 10% CH_3_CH_2_OH-HClO_4_ solution at 20 V. The orientation information of deformed samples was acquired using a Zeiss EVO MA10 scanning electron microscope (SEM) with an Oxford-EBSD detector; particle-free zones (PFZs) were analyzed using the same microscope. Orientation data were analyzed using HKL Channel 5 software (5.11). The foils, with diameters of 3.0 mm after electrolytic thinning, were used for the high-angle annular dark-field scanning transmission electron microscope (HAADF-STEM) observation, using a FEI Titan G2 60-300 (FEI, Hillsboro, OR, USA).

## 3. Results

### 3.1. Deformation Behavior

Figure 1a displays the true stress–true strain curves at 490 °C and an initial strain rate range between 1 × 10^−4^ s^−1^ and 1 × 10^−3^ s^−1^. Based on the resulting change in the true stress–true strain curves, the strain rate sensitivity index *m* can be calculated as follows:(7)$$m=\frac{\partial \mathrm{In}\sigma }{\partial \mathrm{In}\dot{\varepsilon }}$$

The elongation and m-index as a function of the strain rate is shown in Figure 1b. The stress value at the steady state increased from 4.0 to 11.0 MPa, with increasing initial strain rate. The maximum elongation (470%) was obtained when the initial strain rate was 2 × 10^−4^ s^−1^, and the corresponding m-value was about 0.5. Figure 2 shows the grain dynamic and static grain growth behaviors in both the longitudinal (L) and the transverse (T) directions during superplastic deformation and simultaneous static annealing at the same temperature. The grain size increased during superplastic deformation, especially in the longitudinal direction, which nearly doubled from 10.9 μm to 19.9 μm, with a true deformation strain of e = 1.5 compared with the non-deformation samples, and the grain size in the transverse direction increased from 8.1 μm to 10.3 μm. Therefore, the grain was elongated along the tensile direction and the aspect ratio of the grain increased from 1.4 to 2.0. However, the grain size of the static annealing sample only increased slightly in the initial stage, and then stabilized in both directions. This indicates that the grain had good thermal stability under static annealing.

### 3.2. Microstructural Evolution

Figure 3 shows the EBSD orientation images and {110} pole figures after superplastic deformation at 490 °C and 2 × 10^−4^ s^−1^. In the orientation images, low-angle boundaries (LABs, misorientation < 15°) and high-angle boundaries (HABs) are marked by white and black lines, respectively. The samples before deformation had completely recrystallized nearly equiaxed grains. There was no obvious optimal distribution direction of the LABs before deformation. However, most of the LABs were distributed along the transverse direction and the elongated grains were divided during superplastic deformation. A very weak texture was found in undeformed samples and was completely destroyed at a strain of 0.26, which is related to the random rotation of grains during superplastic deformation [26]. There was no obvious change in the misorientation distribution of the grain boundaries, and the proportion of LABs was maintained at about 10%, as shown in Figure 4.

The diffusion creep in superplastic deformation often leads to the occurrence of particle-free zones (PFZs) within the transverse grain boundaries. The obvious PFZs (Figure 5b–f) were observed by SEM when the strain was greater than 0.53, and when the strain was less than 0.99; PFZs were mainly distributed along the transverse grain boundaries. Moreover, PFZs also appeared along the longitudinal grain boundaries when the strain was greater than 0.99, which may have been the result of grain rotation during superplastic deformation. When the strain continued to increase to 1.50 (Figure 5e), some grains composed entirely of PFZs appeared.

The evolution of the HAADF-STEM structure during superplastic deformation is shown in Figure 6. Figure 6c shows that when the true strain was 0.41, a small number of dislocations occurred inside some grains. When the strain increased to 0.77, the dislocation wall appeared inside some grains (Figure 6d, yellow arrow). At the same time, even if it entered a steady stage of superplastic deformation, dislocations were not universal, but rather, only existed in a small number of grains. As shown in Figure 6e, there was a fine grain at the boundary junction. From the magnification of this region (Figure 6f), it can be seen that there was no dislocation in the inner part of this grain while there were many dislocations in the inner part of the neighboring grain.

### 3.3. Superplastic Deformation Mechanisms

The surface study was carried out at a temperature of 490 °C and an initial strain rate of 2 × 10^−4^ s^−1^. The local coarse grids used to observe the offsets in the grain boundaries caused by GBS are shown in Figure 7. Grain neighbor switching was directly observed when the strain increased from 0.18 to 0.54 (Figure 7a–c). In the case that the grains were not obviously elongated, the original separated grains (grains A and B, E and F) became adjacent and the original adjacent grains (grains C and D, G and H) became separated, which is similar to the results observed by Mikhaylovskaya in the 7475 alloy [27]. Obvious offsets were found in the grid lines at some grain boundaries, and some grains rotated significantly during the deformation process. Striated regions at the transverse grain boundary were produced during the entire deformation process and gradually widened as the strain increased, which is consistent with the results observed in [20].

At the primary stage of superplastic deformation, the fine grids within the grains were not deformed and still maintained their original shape and size (Figure 8a,b). However, obvious intragranular deformations occurred in several grains (Figure 8c–f), inside which the dimensions of the fine grids parallel to the tensile principal stress became longer, their dimensions parallel to the compressive principal stress became compressed, and the number of deformed grains increased with the increase in strain (Figure 8e).

The strains caused by grain boundary sliding and intragranular dislocation slip were calculated using the offsets of the coarse grids and the elongation of the intragranular fine grids. The total contribution of grain boundary sliding and intragranular dislocation slip to the superplastic deformation is shown in Table 2. Throughout the whole superplastic deformation stage, the contribution of GBS was 26.2–34.0%, and decreased with increasing strain. The contribution of IDS to the deformation was less than 6.1%. Because superplastic deformation can be explained by three deformation mechanisms—GBS, IDS, and DC—the contribution of DC to deformation was at least 64.0%.

## 4. Discussion

Obvious dynamic grain growth occurred in some alloys during superplastic deformation, which may have been caused by diffusion creep and intragranular dislocation slip, while grain boundary sliding left the grain size unchanged. Significant dynamic grain growth was also found in the AA2050 alloy, with grain size extending to 1.8 times in the longitudinal direction and 1.3 times in the transverse direction after a strain of 1.5. According to the calculation results in Table 2, the contribution of intragranular dislocation slip in the process of superplastic deformation was less than 6.1%, indicating that intragranular dislocation slip was not the main cause of grain growth. It can be inferred that the grain growth of the AA2050 alloy was mainly caused by diffusion creep. The material near the longitudinal grain boundary was transferred to the transverse grain boundary by diffusion, which led to increased longitudinal grain size, while the increase of the transverse grain size may have been caused by the grain rotation, which was consistent with the particle-free zones (Figure 5d,f) near the longitudinal grain boundary. In classical diffusion creep, the grain elongation should be consistent with the macroscopic elongation of the sample, but in the AA2050 alloy, although diffusion creep was the main deformation mechanism, the grain elongation was significantly less than the macroscopic elongation of the sample, which may have been due to grain boundary sliding deformation mechanisms that kept the grains equiaxed during deformation. In the process of superplastic deformation, low-angle boundaries were mostly positioned perpendicular to the tensile axis and the elongated grains were divided into several parts, which may have been because these elongated grains were difficult to coordinate during deformation and were prone to stress concentration, leading to the generation of dislocations. Dislocations formed dislocation walls through slip and climbing, and thus the elongated grains were divided. In addition, fine recrystallized grains were observed at the intersection of grain boundaries, which indicated that dynamic recrystallization occurred during superplastic deformation (the same phenomenon was also observed in [28,29,30]). However, both transmission photographs and surface studies showed that the dislocations played a very limited role in the deformation process, so there remains the question of how these recrystallized grains were produced under the conditions of weak dislocations. After careful observation of the recrystallized grain in Figure 6f, it was found that the interior of the fine recrystallized grain was not only a dislocation-free region, but also a particle-free region, indicating that the formation of the recrystallized grain was related to diffusion creep. It may be that the nearby material diffused through the grain boundaries and deposited at the grain junction to form new grains, which was also a way for the grains to keep equiaxed.

The contribution of diffusion creep to superplastic deformation was more than 64%, indicating that diffusion creep was the main deformation mechanism of the AA2050 alloy. However, there were two kinds of diffusion creep, one being the Herring–Nabarro type (bulk diffusion) [31], and the other the Coble type (grain boundary diffusion) [32]. If the material transfer was carried out through bulk diffusion, the fine grids within the grains would no longer maintain their original shape, and the grid spacing perpendicular to the tensile direction would increase, while the grid spacing perpendicular to the compression direction would decrease. Figure 8 shows that the fine grids within the grains were not obviously deformed, indicating that the deformation of the AA2050 alloy through bulk diffusion was not the main mechanism. In addition, Figure 8 shows that the longitudinal grain boundary grid element disappeared and a striped region was formed at the transverse grain boundary, which indicates that a large number of materials were transferred along the grain boundary through diffusion, which is consistent with the study by Todd [20]. The above evidence shows that within the whole process of superplastic deformation, diffusion creep was carried out through grain boundary diffusion.

Grain boundary sliding and diffusion creep occurred at the same time, and their contribution to superplastic deformation was more than 90.0%. The maximum contribution of intragranular dislocation slip was 6.1%, indicating that dislocation climbing and slip only occurred in areas where local deformation was difficult to coordinate, supplementing the mechanism of grain boundary sliding and diffusion creep, especially in the later stage of superplastic deformation when the grains grew more obviously. After significant grain growth, the sliding among grains was more difficult to coordinate, and the decrease in the number of grain boundaries led to the obstruction of the diffusion channel; the effect of diffusion creep was also limited, which finally led to the fracture of the sample. Contrary to the persistent view that grain boundary sliding is the dominant mechanism of superplastic deformation, diffusion creep always dominated the process of superplastic deformation in this investigation, and grain boundary sliding played a coordinating role.

## 5. Conclusions

The study of the evolution of surface and grain of the AA2050 alloy deformed in the tensile test at a temperature of 490 °C and a strain rate of 2 × 10^−4^ s^−1^ led to the following conclusions:

1. Significant dynamic grain growth occurred in both longitudinal and transverse directions during the tensile test, which was due to the diffusion transfer of matter and grain rotation. At the same time, the grain was elongated after a strain of 1.5; the grain aspect ratio increased from 1.4 to 2.0.

2. Striated regions and particle-free zones caused by diffusion creep were found during the entire process of superplastic deformation. Surface studies indicated that grain boundary diffusion was the dominant form of diffusion creep.

3. Quantitative calculation showed that diffusion creep was the dominant mechanism of the superplastic deformation process; grain boundary sliding was involved as a coordination mechanism; the contribution of diffusion creep to deformation was more than 64.0%, and grain boundary sliding was 26.2–34.0%. As the strain increased, the contribution of grain boundary sliding decreased.

## Figures and Tables

**Figure 1 materials-13-02705-f001:**
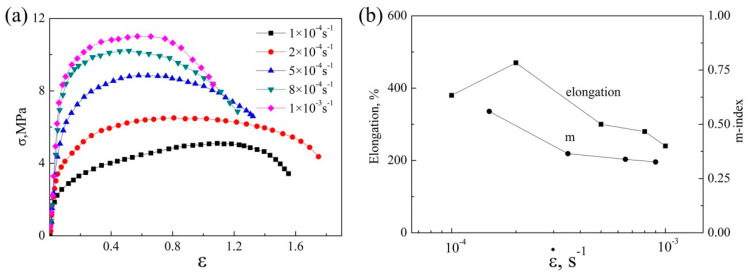
(**a**) The true stress–true strain curves and (**b**) the elongation and m-index at 490 °C of the initial strain rate range of 1 × 10^−4^ s^−1^ to 1 × 10^−3^ s^−1^.

**Figure 2 materials-13-02705-f002:**
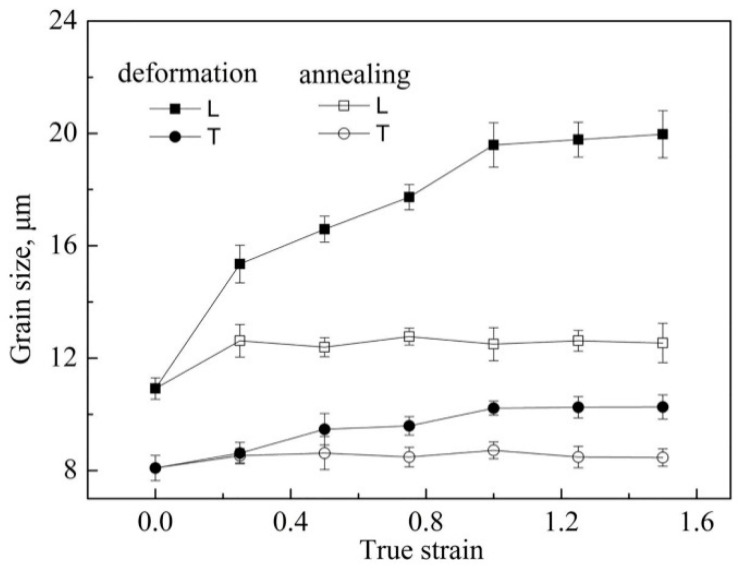
Variation of grain sizes during superplastic deformation and simultaneous annealing at 490 °C and 2 × 10^−4^ s^−1^.

**Figure 3 materials-13-02705-f003:**
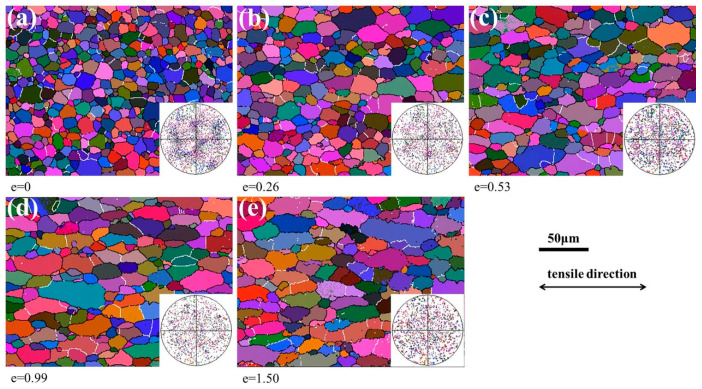
The EBSD orientation images and {110} pole figures after superplastic deformation at 490 °C and 2 × 10^−4^ s^−1^.

**Figure 4 materials-13-02705-f004:**
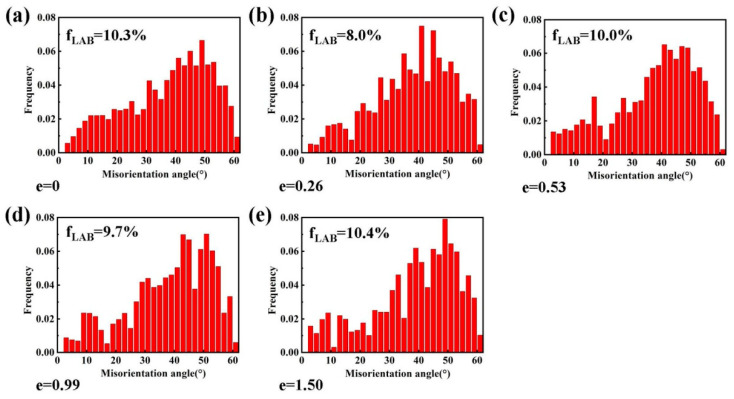
The grain boundary misorientation distribution images after superplastic deformation at 490 °C and 2 × 10^−4^ s^−1^.

**Figure 5 materials-13-02705-f005:**
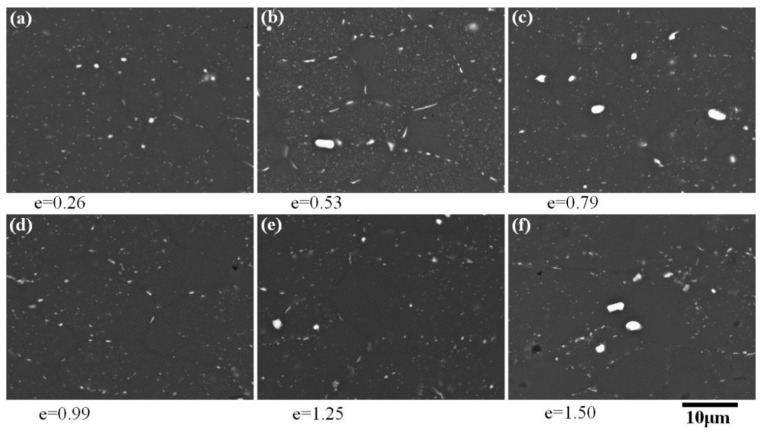
SEM backscattered electron images after superplastic deformation at 490 °C and 2 × 10^−4^ s^−1^.

**Figure 6 materials-13-02705-f006:**
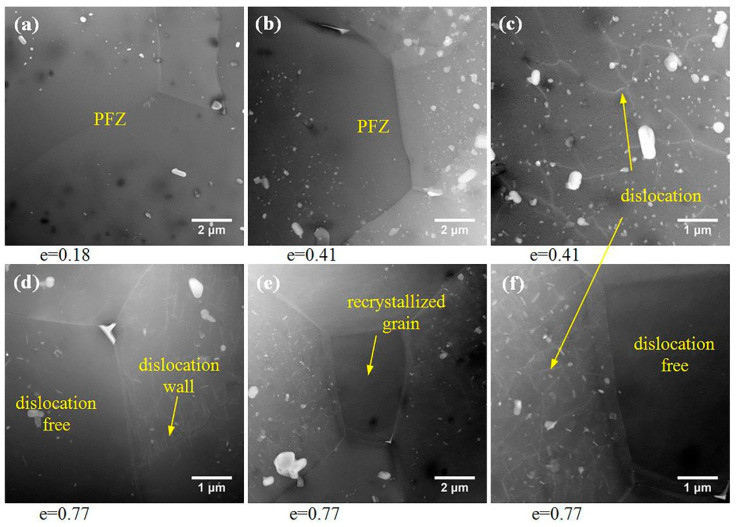
The evolution of the HAADF-STEM structure during superplastic deformation at 490 °C and 2 × 10^−4^ s^−1^. PFZ: particle-free zone.

**Figure 7 materials-13-02705-f007:**
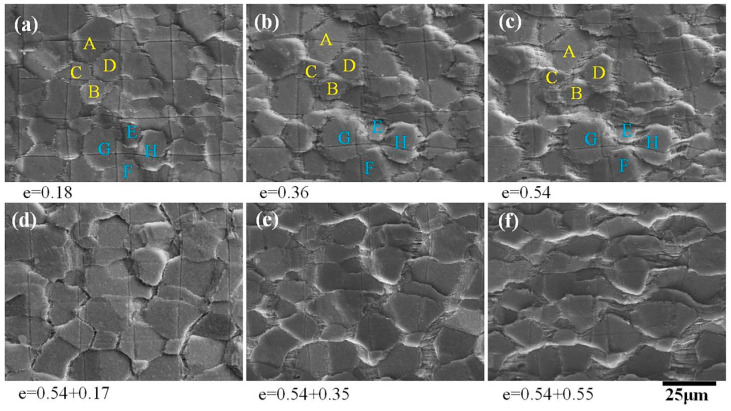
SEM secondary electron images of the local coarse grids after superplastic deformation at 490 °C and 2 × 10^−4^ s^−1^.

**Figure 8 materials-13-02705-f008:**
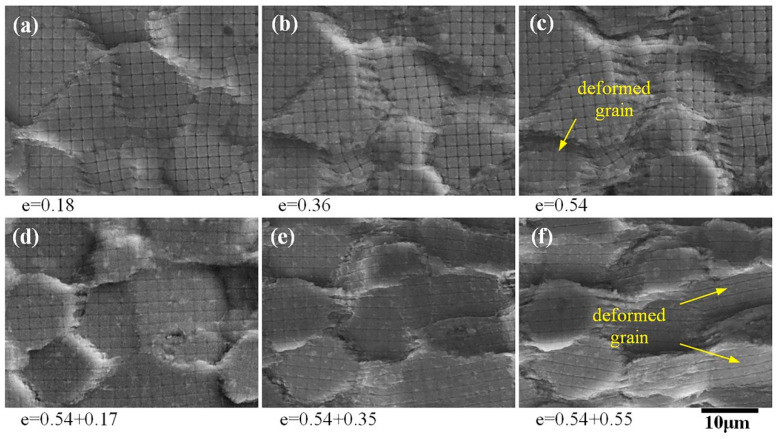
SEM secondary electron images of the local fine grids after superplastic deformation at 490 °C and 2 × 10^−4^ s^−1^.

**Table 1 materials-13-02705-t001:** The composition (in wt.%) of the 2050 alloy.

Cu	Li	Mg	Ag	Mn	Zr	Ti	Zn	Fe	Si	Al
3.24	0.83	0.32	0.35	0.38	0.083	0.038	0.053	0.045	0.053	bal.

**Table 2 materials-13-02705-t002:** Contributions of grain boundary sliding (GBS) and intragranular dislocation slip (IDS) to the total strain during superplastic deformation at 490 °C and 2 × 10^−4^ s^−1^.

True Strain	Longitudinal Strain ε, %	Intragranular Dislocation Slip		Transverse Strain ε^tr^, %	Grain Boundary Sliding	
		IDS Strain ε_IDS_, %	Contribution to Total Strain γ_GBS_, %		GBS Strain $\varepsilon_{\mathrm{GBS}}^{\mathrm{tr}}$, %	Contribution to Total Strain γ_GBS_, %
0.18	19.6	0.40 ± 0.17	2.0	9.1	3.11 ± 0.58	34.0
0.36	44.0	1.13 ± 0.84	2.6	15.3	5.09 ± 1.04	33.4
0.54	71.3	2.44 ± 1.69	3.4	21.8	6.77 ± 1.17	31.0
0.54 + 0.17	18.9	1.16 ± 0.75	6.1	9.0	2.50 ± 0.69	27.7
0.54 + 0.35	41.5	2.50 ± 1.19	6.0	16.1	4.29 ± 1.78	26.7
0.54 + 0.55	72.8	3.66 ± 2.54	5.0	22.2	5.81 ± 2.23	26.2

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
