# Peer review of "Superplastic Deformation Mechanisms in Fine-Grained 2050 Al-Cu-Li Alloys"

_materials, 2020, doi:10.3390/ma13122705_

Round 1
Reviewer 1 Report
The paper is generally acceptable. There aren’t many errors and errors in logic interpretation of obtained results. The abstracts and conclusions are sufficiently concise and clear. Generally, the paper is well written.
However, there are several issues that must be clarified or corrected before the article will be suitable for publication:
- Introduction part. Please explain the motivation of choice Al-Cu-Li alloys in this article?
- 6. What kind of particles (phase composition) are visible in Fig. 6. Only Al7Cu4Li phase?
- Dislocation configurations in Fig. 6 are not well visible. Dark Field (DF) or ZC contrast in STEM are not the best for imaging of dislocation structure. In Fig. 6- please complete the STEM imaging type.
- Page 3 line 111. “… were used for the transmission electron (TEM) observation". Where are the results of microstructure investigation with using (TEM)?
- minor language errors:
- page 3 line 111, “transmission electron (TEM)” should be - transmission electron microscope
- page 2 line 82, FIB instrument- Focusing Ion Beam (FIB)
- page 4 line 136, LAB should be LABs
- page 4 line 138, the sentence “the sample before deformation have complete recrystalization nearly equiaxed grains” maybe- the sample before deformation have completly recrystalized nearly equiaxed grains
- page 8 line 209, the sentence “significant dynamic grain dynamic growth” maybe significant dynamic grain growth
I can recommend publication of the manuscript for publication after minor revision.
Reviewer 2 Report
Review for Materials- 822892
Superplastic deformation mechanisms in fine-grained 2050 Al-Cu-Li alloys
The authors address an interesting research topic for the journal Materials. It is a well-organized paper but some recommendations should be considered for its publication:
- Please explain extensively how you can get figures 8 and 9. SEM was used during the deformation? This question should be clarified in the manuscript.
- A more extensive state-of-the-art could be included in this study.
Other minor Changes
- For the sake of clarity, define m-index in the manuscript
- All acronyms should be defined the first time they appear.
- Font size should be increased in the graphs of Figure 4 (axis, numbers, etc.)
- Figure 4. For better comparison, unify the vertical axis in all the graphs.
Reviewer 3 Report
The superplasticity of an Al alloy was investigated from the viewpoints of the change in morphology as well as the change in microstructure. Based on these results, the evaluation of the effects of dislocations, creep and grain boundary sliding was conducted. In spite that the temperature was limited to a particular temperature, the results given by the authors may explore the deformation mechanism at different temperatures.
The manuscript is easily understood to me.
I think, this paper is acceptable for publication.
Author Response
Thank you very much for you appreciation.
Reviewer 4 Report
The paper is of significant interest, although performed on some ordinary materials, since superplastic deformation has considerable potential for manufacturing industry. Research methodology is focused on microstructural aspects, which is meaningful for understanding material behavior. Some advanced investigations methods are used, while discussions and conclusions are well formulated. Good job!
Author Response
Thank you very much for your appreciation.
Round 2
Reviewer 2 Report
Review for Materials- 822892
Superplastic deformation mechanisms in fine-grained 2050 Al-Cu-Li alloys
The explanation for obtaining Figures 7 and 8 follows the logical process. Anyway, the precision of obtaining the same image with a different deformation level is surprising. It could be said that it can generate some mistrust if an explanation of how such precision can be obtained is not included in the manuscript: how can the authors locate the same area (approximately 40 µm) over and over again?
